



# Geochemical characterisation and protolith restoration of metamorphic rocks at Lazishao graphite mine, Sichuan

**Wenqi Cheng [2], Haijun Yu[1,2]\*, Xue Wang [3], Decai Kong [2], Bo Long [2]**

1.Sichuan Shu-Neng Mineral Co., Ltd., Leshan,Sichuan, 614600,China;
2. Mine bureau of sichuan province,106 geological team, chengdu,Sichuan, 611130,China;
3. Sichuan Shu-Dao Highway Group Co., Ltd. Chengdu,Sichuan,610000, China.
\* Correspondence: yuhaijun0601@163.com

**Abstract:** This study determined the deposit characteristics and geochemical features of metamorphic rocks from the Lazishao graphite deposit in order to reconstruct the metamorphic protoliths and palaeo-sedimentary environment. The results show that the $SiO_2$ content of the metamorphic rocks is high (55.60% to 77.94%), while $Na_2O$ is 0.22% to 1.85%, $K_2O$ is 1.87% to 3.45%, $K_2O > Na_2O$, and $K_2O/Na_2O + K_2O > 0.5$. The fractionation degree of light rare earth elements (LREEs) is greater than that of heavy rare earth elements (HREEs), with LREE/HREE ratios of 3.09 to 8.77; $La_N/Yb_N$ is 2.72 to 10.75, with a mean value of 9.69. The rocks have moderate negative Eu anomalies (δEu = 0.50 to 0.89, mean = 0.64). Ionic lithophile elements (e.g., Rb, Ba, and K) are relatively enriched, but Sr is relatively depleted. The graphite-bearing metamorphic rocks in the study area originated from sedimentary rocks, mainly mudstone and greywacke. The palaeo-sedimentary environment was a low-salinity terrestrial freshwater body in a cold or moderately cold climatic zone.

**Keywords:** graphite mine; deposit characteristics; geochemical characteristics; carbon source; Lazishao

## 1. Introduction

Graphite (also known as 'black gold') is one of China's strategic non-metallic mineral resources(Deng ShaoJun, 2020). Graphite has electrical and thermal conductivities similar to those of metallic materials, and desirable plasticity and expandability (Zhang TengFei, 2015). It is widely used in various industrial fields, including metallurgy, mechanics, chemistry, and electricity, and has become a strategic resource for modern cutting-edge technologies (Zhang et al., 2013; Li et al., 2015; Jiang , 2016; Wang et al., 2017).

Graphite mines are widely distributed in China, although those with large-scale output are mainly concentrated in five regions: Shandong, Heilongjiang, Hunan, Inner Mongolia, and Sichuan. Graphite mines in Sichuan Province are mainly distributed in Nanjiang County of Bazhong City and Renhe District and Yanbian County of Panzhihua City. The national resource base (Nanjiang–Wangcang graphite mine) is in Sichuan Province and the graphite mineral resource exploration and development base is in Panzhihua City (Yu et al., 2017; Department of Natural Resources of Sichuan Province,2017). At present, ultra-large graphite deposits at Zhongba and Tianping in Panzhihua City have been discovered. The metallogenic age of graphite deposits in Panzhihua is mainly Palaeoproterozoic–Mesoproterozoic, and metallogenic processes included the deposition of graphitic rocks, regional metamorphism, and late-stage superposed contact metamorphism (Yu et al., 2020).

However, the microscopic characteristics, geochemistry, ore genesis, carbon source, and other deposit features of graphite mines in the region have not been thoroughly explored. In this study, we investigated the geochemistry of metamorphic rocks from the Lazishao graphite mine (Renhe District, Panzhihua City) in order to determine the metamorphic protoliths and palaeo-sedimentary environment. The results of this study provide a reference for analysing the metallogenic mechanism and genesis of sedimentary–metamorphic graphite deposits in the Panxi area of Sichuan Province and across China.





## 2. Regional geological background

*2.1. Geotectonic position*

The study area is in the central Yangzi Craton (Kangdian axis), east of the Songpan–Ganzi orogenic belt, and at the northeast tip of the Gondwana palaeo-continent. This region forms an important part of the Panxi metallogenic belt (Fig. 1a). The Kangdian basement fault uplift zone is a horst-like structure composed of Archaean to Early Mesozoic metamorphosed magmatic complexes with a banded distribution. Magmatic rocks are primarily Jinning granites of the Chengjiang period, whose extension is controlled by north–south trending faults. The primary metallogenic belt is the Fe–Cu–V–Ti–Ni–Sn–Pb–Zn–Au–Pt–rare earth–asbestos metallogenic belt along the Yangtze metallogenic province of the Kangdian fault uplift zone; the secondary metallogenic belt is the Cu–Ni–Pb–Zn–graphite sub-metallogenic belt of the Yanbian palaeo-forearc basin.

*2.2. Geological characteristics of the mining area*

The regional stratigraphy is simple, mainly consisting of the second member of the Lengzhuguan Formation (Kangding Group), the first member of the Neoproterozoic Guanyinya Formation, and the first member of the Cenozoic Yuanyongjing Formation. The second member of the Lengzhuguan Formation is the ore-bearing formation of the graphite deposits, whose lithology is mainly sericite (muscovite)–quartz schist and two-mica–quartz schist. During the massive intrusion of monzonitic granites in the Cryogenian, most schists of the Lengzhuguan Formation were removed, leaving only a small number of lenticular and strip-shaped outcrops of the Lengzhuguan Formation in the west(Fig. 1b).

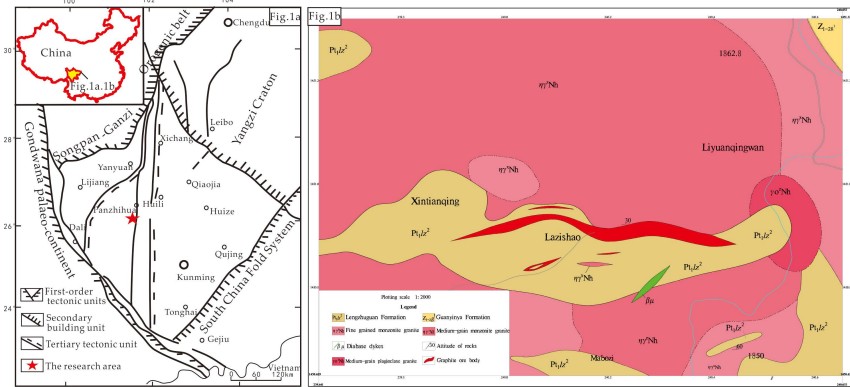

Fig.1 Geotectonic location and geological map of the research area(Wang Xue , 2014)

## 3. Petrographic characteristics

*3.1. Ore types*

Natural graphite deposits at Lazishao are mainly sericite–quartz schist; some graphite infills between quartz grains and some is arranged parallel to mica and quartz flakes, showing a schistose texture. Graphite flake sizes are 48–680 μm and the ore is a crystalline flake graphite.

*3.2. Ore characteristics*

The ore mineral is graphite; gangue minerals are mainly quartz, muscovite, and sericite, with occasional biotite and feldspar; metal minerals mainly include magnetite, hematite, limonite, pyrrhotite, and pyrite. Fresh graphite surfaces are black, and weathered surfaces are brownish-black. Granular minerals such as quartz (anhedral) and a small amount of feldspar (anhedral to subhedral) are distributed in the ore, while flaky minerals, including graphite and mica, have a directional arrangement among the granular minerals. Flaky minerals are less abundant than granular minerals, exhibiting a grano-lepidoblastic texture with a predominantly schistose structure (Fig. 2 and Fig. 3) following the banded structure. The graphite content is about 10% and mostly comprises individual flakes or flake aggregates. Graphite particles are



generally 0.2 to 0.6 mm in length and 0.05 to 0.1 mm in width. Graphite is mostly in banded and
directional, extending along the same direction as muscovite and biotite, and graphite grains
are evenly distributed among the quartz grains (Fig. 4). Euhedral columnar minerals are
occasionally seen. The quartz content is ~60%, and particles are anhedral and granular with a
small grain size (< 0.8 mm), showing a mosaic structure. Quartz grains have been deformed
under stress, with flattened and elongated morphology, displaying wavy extinction and optical
anomalies; however, the long axis of quartz grains is still in a directional arrangement with
flaky minerals. Mica is mostly muscovite and sericite, with a small amount of biotite. Muscovite
and sericite (flake diameters 0.05 to 1.9 mm mm) have a silky lustre and typically represent 20%
of the sample; the content of muscovite is higher than that of sericite. The biotite content is
relatively low (generally 2% to 5%) and grains are unevenly distributed. Biotite particles are
usually brown and flaky, generally 0.25 to 0.6 mm in diameter, and have a directional
arrangement. Metallic minerals account for 0.1% to 3% and are unevenly distributed in the ore.
Magnetite accounts for ~90% of the metallic minerals, followed by pyrite, with occasional
arsenopyrite and chalcopyrite. These minerals are generally interstitial between the particles of
major minerals.

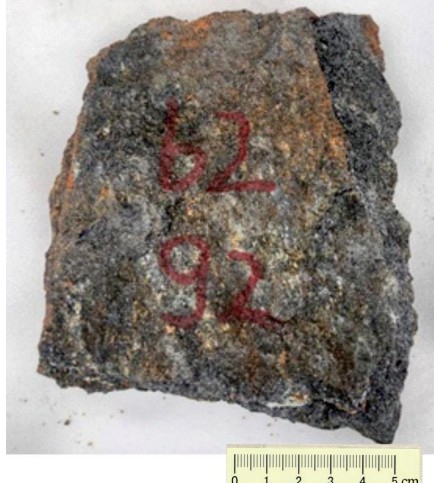
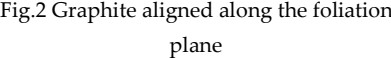

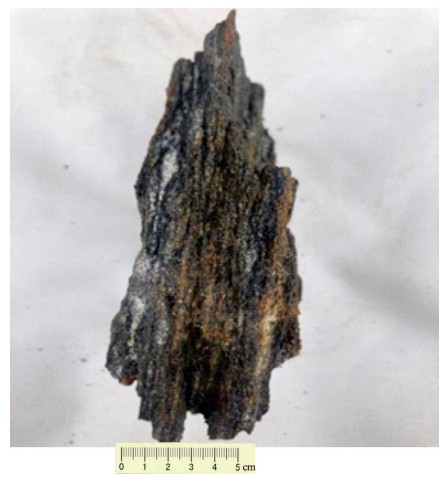

Fig.2 Graphite aligned along the foliation plane

Fig.3 Flake structure of graphite ore


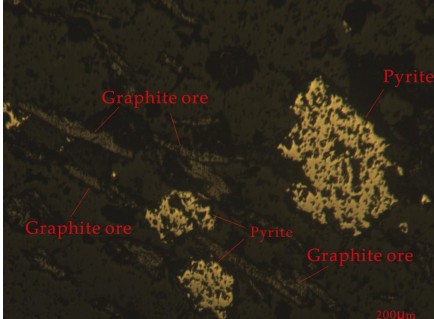

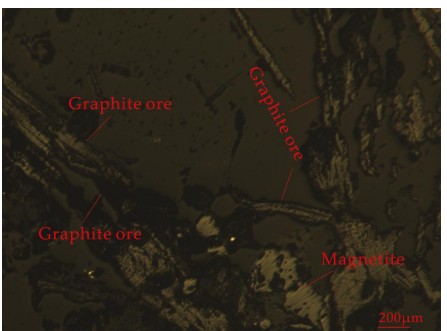

Fig.4 Oriented Graphite ore (polished section10×10)

*3.3. Ore-fixed carbon content*





The fixed carbon content of graphite ores in the study area ranges from 2.03% to 18.87%,
and the average fixed carbon content of industrial-grade ores in the mining area is 5.16%, with
the fixed carbon occurring in graphite. The fixed carbon content of muscovite (sericite)–quartz
schist is mainly ~1%. There is little fixed carbon in the of granites and gangue rocks, and the
distribution of fixed carbon content in the deposit is irregular.
*3.4. Graphite flake size*
Graphite flake size determines the quality of graphite products. Based on the analysis of 22
samples, graphite flakes in the mining area are generally between 0.048 and 0.68 mm. Graphite
flakes of +100 mesh account for 17% to 80%, with an average of 52.4%; 100 to 80 mesh graphite
flakes account for 9% to 28%, with an average of 19.4%; 80 to 50 mesh graphite flakes account
for 5% to 35%, with an average of 19.8%; graphite flakes of > 50 mesh account for 3% to 38%,
with an average of 13.3%. Graphite flake size increases with depth, making the deep orebody
more favourable than the surface orebody.
**4. Materials and methods**
A total of 11 fresh samples were collected from drill holes, including seven graphite ore
samples and four mica–quartz schist samples. Samples were analysed at the Mineral Testing
Centre of Xichang, Sichuan Provincial Bureau of Geology and Mineral Exploration and
Development (Table 1).
Table 1    Analysis of Lazishao Graphite Mine Samples

| Analytical object | Analytical method | Analytical accuracy |
|---|---|---|
| Major elements | X-ray fluorescence spectrometry | Better than 0.1% to 1.0% |
| $V_2O_5$ and $Fe_2O_3$ | Inductively coupled plasma–atomic emission spectrometry | Reproducibility up to 5% |
| Trace elements and rare earth elements | Inductively coupled plasma–mass spectrometry (ICP–MS) | Better than 10% |

**5. Results**
*5.1. Geochemical characteristics of major elements*
Major elements compositions of the graphite ore and mica–quartz schist are shown in
Table 2. The $SiO_2$ content is generally high, ranging from 55.60% to 77.94%, with an average of
68.74%, which is higher than the average $SiO_2$ content in the upper crust (66%) (Deng ShaoJun,
et al.,2020). $Na_2O$ show little variation, ranging from 0.22% to 1.85%, with an average of 0.68%,
while $K_2O$ ranges from 1.87% to 3.45%, with an average of 2.70%. In all samples, $K_2O > Na_2O$,
and $K_2O/Na_2O + K_2O > 0.5$ (4.64–12.31, with an average of 8.32), indicating that the protoliths of
the graphite deposit and mica–quartz schist were of normal sedimentary origin (He Tongxing
et al.,1980). $TiO_2$ ranges from 0.14% to 1.36% (average of 0.44%), MgO ranges from 0.59% to
5.11%, and CaO ranges from 0.14% to 3.22%. In all samples, CaO < MgO, which also indicates
that the metamorphic protoliths had a normal sedimentary origin (He Tongxing et al.,1980).
The $Fe_2O_3$ content is 2.43% to 3.92%, with an average of 2.99%; FeO ranges from 0.50% to 5.50%,
with an average value of 2.52; $Al_2O_3$ ranges from 5.45% to 14.81%, with an average of 9.96%. The
$SiO_2/Al_2O_3$ ratio is between 3.75 and 13.84 (7.96–13.84 for the seven graphite ore samples and
3.75–4.69 for the four mica–quartz schist samples), with an average of 7.85%. This shows that
the maturity levels of the seven graphite ore samples are similar, and those of the four
mica–quartz schist samples are similar (Feng Wei, 2019). $SiO_2$ is significantly negatively





correlated with $Al_2O_3$. $P_2O_5$ ranges from 0.072% to 0.92%, which is generally low, and MnO is
between 0.010% and 0.090%, with a small variation range.
140        On Harker diagrams (Fig. 5), $SiO_2$ is negatively correlated with $Al_2O_3$, $Na_2O$, $K_2O$, $TiO_2$,
CaO, MnO, MgO, and $Fe_2O_3$, and positively correlated with $P_2O_5$ and $V_2O$. On this basis, the
chemical differentiation of the rocks is constrained by sedimentary differentiation (He et
al.,1980; Long , 2016).

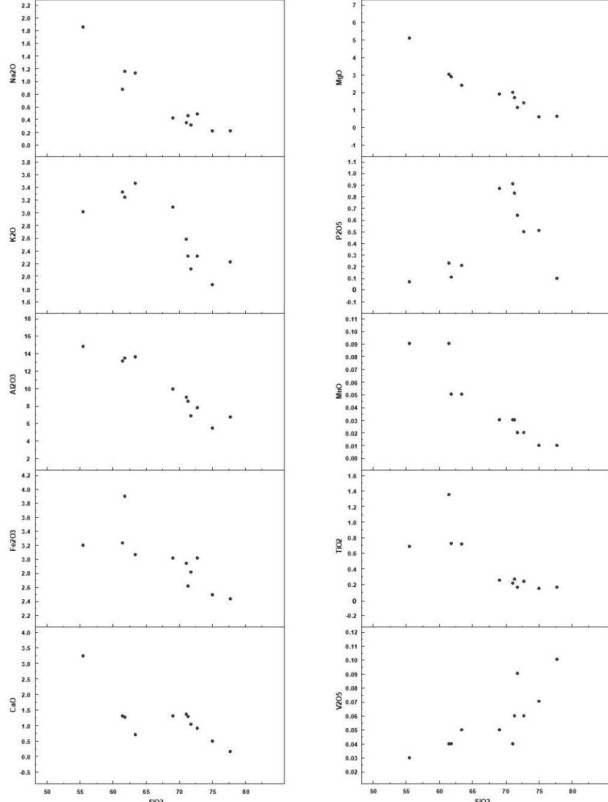

Fig.5 Harker diagrams for the Lazishao Graphite mine




Table 2 Results of major elements analysis of Lazishao graphite deposit metamorphic rocks(*wt.%*)

| sample number | sample type | $Na_2O$ | $K_2O$ | $SiO_2$ | $Al_2O_3$ | $Fe_2O_3$ | $CaO$ | $MgO$ | $MnO$ | $V_2O_5$ | $TiO_2$ | $P_2O_5$ | $FeO$ | $H_2O^+$ | loss on ignition | TOTAL |
|---|---|---|---|---|---|---|---|---|---|---|---|---|---|---|---|---|
| LZS-BZY-01 | mica–quartz schist sample | 1.85 | 3.01 | 55.60 | 14.81 | 3.20 | 3.22 | 5.11 | 0.087 | 0.035 | 0.68 | 0.072 | 5.09 | 2.58 | 4.77 | 100.11 |
| LZS-BZY-02 | mica–quartz schist sample | 1.16 | 3.26 | 62.21 | 13.49 | 3.92 | 1.27 | 2.89 | 0.046 | 0.043 | 0.72 | 0.11 | 4.00 | 2.00 | 5.38 | 100.50 |
| LZS-SMK-01 | graphite ore sample | 0.34 | 2.61 | 72.00 | 9.05 | 2.97 | 1.37 | 2.00 | 0.027 | 0.039 | 0.21 | 0.92 | 1.50 | 1.42 | 6.72 | 101.18 |
| LZS-SMK-02 | graphite ore sample | 0.48 | 2.33 | 73.50 | 7.84 | 3.04 | 0.91 | 1.40 | 0.023 | 0.062 | 0.23 | 0.50 | 1.33 | 0.70 | 8.60 | 100.95 |
| LZS-SMK-03 | graphite ore sample | 0.22 | 2.22 | 77.94 | 6.72 | 2.43 | 0.14 | 0.61 | 0.010 | 0.10 | 0.16 | 0.10 | 0.59 | 0.96 | 7.98 | 100.18 |
| LZS-SMK-04 | graphite ore sample | 0.22 | 1.87 | 75.41 | 5.45 | 2.50 | 0.48 | 0.59 | 0.013 | 0.071 | 0.14 | 0.51 | 0.50 | 0.70 | 12.02 | 100.47 |
| LZS-SMK-05 | graphite ore sample | 0.42 | 3.10 | 69.63 | 9.98 | 3.03 | 1.31 | 1.91 | 0.026 | 0.048 | 0.25 | 0.88 | 1.33 | 1.38 | 7.40 | 100.69 |
| LZS-SMK-06 | graphite ore sample | 0.31 | 2.12 | 72.28 | 6.90 | 2.83 | 1.04 | 1.14 | 0.021 | 0.090 | 0.16 | 0.64 | 1.42 | 1.00 | 10.64 | 100.59 |
| LZS-SMK-07 | graphite ore sample | 0.45 | 2.33 | 72.19 | 8.56 | 2.64 | 1.29 | 1.72 | 0.027 | 0.058 | 0.26 | 0.84 | 1.92 | 1.40 | 7.38 | 101.07 |
| LZS-BZY-03 | mica–quartz schist sample | 1.12 | 3.45 | 63.18 | 13.53 | 3.05 | 0.69 | 2.37 | 0.049 | 0.046 | 0.71 | 0.21 | 4.59 | 1.94 | 4.68 | 99.62 |
| LZS-BZY-04 | mica–quartz schist sample | 0.88 | 3.36 | 62.16 | 13.25 | 3.27 | 1.30 | 3.05 | 0.090 | 0.039 | 1.36 | 0.23 | 5.50 | 2.16 | 4.44 | 101.09 |

Table 3 Rare earth elements analysis results of Lazishao graphite deposit metamorphic rocks andrelevant parameter value(ppm)

| sample number | sample type | La | Ce | Pr | Nd | Sm | Eu | Gd | Tb | Dy | Ho | Er | Tm | Yb | Lu | ΣREE | LREE | HREE | LREE/HREE | $La_N/Yb_N$ | δEu | δCe |
|---|---|---|---|---|---|---|---|---|---|---|---|---|---|---|---|---|---|---|---|---|---|---|
| LZS-BZY-01 | mica–quartz schist sample | 36.6 | 70.1 | 8.64 | 32.6 | 6.36 | 1.33 | 4.49 | 0.96 | 6.68 | 1.62 | 4.31 | 0.71 | 4.09 | 0.64 | 179.13 | 155.63 | 23.50 | 6.62 | 6.42 | 0.72 | 0.93 |
| LZS-BZY-02 | mica–quartz schist sample | 61.1 | 114 | 14.3 | 51.6 | 9.87 | 1.69 | 6.94 | 1.44 | 9.90 | 2.43 | 6.12 | 1.01 | 5.58 | 0.85 | 286.83 | 252.56 | 34.27 | 7.37 | 7.85 | 0.59 | 0.91 |
| LZS-SMK-01 | graphite ore sample | 35.6 | 88.3 | 10.3 | 41.0 | 9.54 | 1.59 | 7.43 | 1.69 | 12.4 | 3.24 | 8.5 | 1.41 | 8.40 | 1.32 | 230.72 | 186.33 | 44.39 | 4.20 | 3.04 | 0.56 | 1.12 |
| LZS-SMK-02 | graphite ore sample | 26.9 | 36.9 | 7.52 | 31.8 | 7.44 | 1.30 | 6.10 | 1.37 | 10.0 | 2.65 | 6.8 | 1.12 | 7.10 | 1.06 | 148.06 | 111.86 | 36.20 | 3.09 | 2.72 | 0.57 | 0.63 |
| LZS-SMK-03 | graphite ore sample | 19.7 | 27.3 | 5.06 | 20.5 | 4.45 | 1.13 | 2.99 | 0.59 | 4.65 | 1.09 | 2.80 | 0.56 | 3.32 | 0.50 | 94.64 | 78.14 | 16.50 | 4.74 | 4.26 | 0.89 | 0.65 |
| LZS-SMK-04 | graphite ore sample | 30.2 | 35.5 | 7.94 | 33.1 | 7.05 | 1.29 | 4.55 | 1.00 | 5.05 | 1.11 | 2.82 | 0.44 | 2.51 | 0.40 | 132.96 | 115.08 | 17.88 | 6.44 | 8.63 | 0.65 | 0.55 |
| LZS-SMK-05 | graphite ore sample | 47.6 | 112 | 13.4 | 52.8 | 11.6 | 2.16 | 8.54 | 1.86 | 12.7 | 3.14 | 7.7 | 1.27 | 7.43 | 1.18 | 283.38 | 239.56 | 43.82 | 5.47 | 4.60 | 0.63 | 1.07 |
| LZS-SMK-06 | graphite ore sample | 27.2 | 39.1 | 8.12 | 34.4 | 8.00 | 1.29 | 6.42 | 1.27 | 8.50 | 2.06 | 4.99 | 0.79 | 4.76 | 0.66 | 147.56 | 118.11 | 29.45 | 4.01 | 4.10 | 0.53 | 0.64 |
| LZS-SMK-07 | graphite ore sample | 40.5 | 82.6 | 11.3 | 45.2 | 9.89 | 1.43 | 7.12 | 1.50 | 10.4 | 2.59 | 6.6 | 1.12 | 6.53 | 1.07 | 227.85 | 190.92 | 36.93 | 5.17 | 4.45 | 0.50 | 0.93 |
| LZS-BZY-03 | mica–quartz schist sample | 56.5 | 106 | 13.3 | 49.7 | 9.48 | 1.76 | 6.43 | 1.26 | 8.01 | 1.81 | 4.46 | 0.68 | 3.77 | 0.57 | 263.73 | 236.74 | 26.99 | 8.77 | 10.75 | 0.65 | 0.92 |
| LZS-BZY-04 | mica–quartz schist sample | 45.4 | 89.0 | 11.5 | 42.8 | 8.76 | 1.91 | 6.24 | 1.29 | 8.71 | 2.12 | 5.55 | 0.84 | 4.69 | 0.71 | 229.52 | 199.37 | 30.15 | 6.61 | 6.94 | 0.75 | 0.93 |




149        Table 4 Trace element analysis results of Lazishao graphite deposit metamorphic rocks(ppm)

| sample number | sample type | Rb | Ba | Th | U | K | Ta | Nb | La | Ce | Sr | Nd | P | Zr | Hf | Sm | Ti | Y |
|---|---|---|---|---|---|---|---|---|---|---|---|---|---|---|---|---|---|---|
| LZS-BZY-01 | mica–quartz schist sample | 162.00 | 620.00 | 10.40 | 1.82 | 25000.00 | 1.47 | 10.50 | 36.60 | 70.10 | 80.80 | 32.60 | 316.80 | 255.00 | 6.11 | 6.36 | 4100.00 | 40.00 |
| LZS-BZY-02 | mica–quartz schist sample | 167.00 | 800.00 | 18.60 | 2.65 | 27100.00 | 1.66 | 14.60 | 61.10 | 114.00 | 40.90 | 51.60 | 484.00 | 276.00 | 6.00 | 9.87 | 4300.00 | 61.60 |
| LZS-SMK-01 | graphite ore sample | 136.00 | 530.00 | 9.62 | 4.02 | 21700.00 | 1.93 | 6.82 | 35.60 | 88.30 | 13.90 | 41.00 | 4048.00 | 267.00 | 6.12 | 9.54 | 1300.00 | 78.20 |
| LZS-SMK-02 | graphite ore sample | 103.00 | 480.00 | 9.47 | 7.56 | 19300.00 | 1.61 | 6.52 | 26.90 | 36.90 | 12.00 | 31.80 | 2200.00 | 141.00 | 3.65 | 7.44 | 1400.00 | 66.70 |
| LZS-SMK-03 | graphite ore sample | 84.10 | 330.00 | 6.15 | 6.27 | 18400.00 | 1.80 | 6.28 | 19.70 | 27.30 | 9.25 | 20.50 | 440.00 | 154.00 | 4.17 | 4.45 | 1000.00 | 26.90 |
| LZS-SMK-04 | graphite ore sample | 70.00 | 420.00 | 6.41 | 4.88 | 15500.00 | 0.38 | 3.34 | 30.20 | 35.50 | 11.10 | 33.10 | 2244.00 | 181.00 | 3.45 | 7.05 | 840.00 | 29.50 |
| LZS-SMK-05 | graphite ore sample | 142.00 | 650.00 | 11.50 | 6.41 | 25700.00 | 0.94 | 8.03 | 47.60 | 112.00 | 27.40 | 52.80 | 3872.00 | 244.00 | 6.50 | 11.60 | 1500.00 | 74.30 |
| LZS-SMK-06 | graphite ore sample | 79.00 | 400.00 | 8.30 | 5.44 | 17600.00 | 0.64 | 4.81 | 27.20 | 39.10 | 8.80 | 34.40 | 2816.00 | 171.00 | 3.41 | 8.00 | 1000.00 | 50.60 |
| LZS-SMK-07 | graphite ore sample | 102.00 | 460.00 | 8.66 | 4.57 | 19300.00 | 1.35 | 8.68 | 40.50 | 82.60 | 17.70 | 45.20 | 3696.00 | 201.00 | 5.51 | 9.89 | 1600.00 | 63.20 |
| LZS-BZY-03 | mica–quartz schist sample | 184.00 | 850.00 | 14.20 | 3.57 | 28600.00 | 10.70 | 17.00 | 56.50 | 106.00 | 56.00 | 49.70 | 924.00 | 314.00 | 4.00 | 9.48 | 4300.00 | 44.90 |
| LZS-BZY-04 | mica–quartz schist sample | 195.00 | 670.00 | 11.50 | 2.83 | 27900.00 | 1.56 | 16.50 | 45.40 | 89.00 | 63.50 | 42.80 | 1012.00 | 253.00 | 3.06 | 8.76 | 8200.00 | 52.10 |
| sample number | sample type | Yb | Lu | Ni | Cr | Co | al | fm | c | alk | si | Zr | Zr/TiO2 | al-alk | La/Th | Rb/Sr | Sr/Ba | / |
| LZS-BZY-01 | mica–quartz schist sample | 4.09 | 0.64 | 48.90 | 322.00 | 25.40 | 14.79 | 16.67 | 3.22 | 4.86 | 25.92 | 255.00 | 375.00 | 9.93 | 3.52 | 2.00 | 0.13 | / |
| LZS-BZY-02 | mica–quartz schist sample | 5.58 | 0.85 | 69.80 | 89.20 | 22.50 | 13.42 | 14.71 | 1.26 | 4.39 | 28.89 | 276.00 | 383.33 | 9.03 | 3.28 | 4.08 | 0.05 | / |
| LZS-SMK-01 | graphite ore sample | 8.40 | 1.32 | 77.40 | 39.90 | 29.70 | 8.94 | 9.37 | 1.35 | 2.92 | 33.21 | 267.00 | 1271.43 | 6.02 | 3.70 | 9.78 | 0.03 | / |
| LZS-SMK-02 | graphite ore sample | 7.10 | 1.06 | 108.00 | 62.30 | 17.50 | 7.77 | 8.75 | 0.90 | 2.79 | 33.98 | 141.00 | 613.04 | 4.98 | 2.84 | 8.58 | 0.03 | / |
| LZS-SMK-03 | graphite ore sample | 3.32 | 0.50 | 28.70 | 51.40 | 17.60 | 6.71 | 6.07 | 0.14 | 2.44 | 36.31 | 154.00 | 962.50 | 4.27 | 3.20 | 9.09 | 0.03 | / |
| LZS-SMK-04 | graphite ore sample | 2.51 | 0.40 | 78.20 | 62.80 | 16.30 | 5.42 | 6.08 | 0.48 | 2.08 | 35.02 | 181.00 | 1292.86 | 3.34 | 4.71 | 6.31 | 0.03 | / |
| LZS-SMK-05 | graphite ore sample | 7.43 | 1.18 | 98.20 | 65.40 | 28.00 | 9.91 | 9.27 | 1.30 | 3.50 | 32.27 | 244.00 | 976.00 | 6.41 | 4.14 | 5.18 | 0.04 | / |
| LZS-SMK-06 | graphite ore sample | 4.76 | 0.66 | 112.00 | 48.80 | 17.10 | 6.86 | 8.18 | 1.03 | 2.42 | 33.53 | 171.00 | 1068.75 | 4.44 | 3.28 | 8.98 | 0.02 | / |
| LZS-SMK-07 | graphite ore sample | 6.53 | 1.07 | 99.20 | 43.00 | 21.00 | 8.47 | 8.85 | 1.28 | 2.76 | 33.33 | 201.00 | 773.08 | 5.71 | 4.68 | 5.76 | 0.04 | / |
| LZS-BZY-03 | mica–quartz schist sample | 3.77 | 0.57 | 63.80 | 92.70 | 23.90 | 13.58 | 13.16 | 0.69 | 4.58 | 29.60 | 314.00 | 442.25 | 9.00 | 3.98 | 3.29 | 0.07 | / |
| LZS-BZY-04 | mica–quartz schist sample | 4.69 | 0.71 | 65.70 | 74.50 | 25.30 | 13.11 | 15.01 | 1.29 | 4.19 | 28.70 | 253.00 | 187.41 | 8.92 | 3.95 | 3.07 | 0.09 | / |






### 5.2. Geochemical characteristics of rare earth elements


Rare earth element (REE) data of the 11 samples are shown in Table 3. Total REE (ΣREE)
ranges from 94.64 to 286.83 ppm, with an average of 202.22 ppm; total light REE (ΣLREE)
ranges from 78.14 to 252.56 μg/g, with an average of 171.3 μg/g; total heavy REE (ΣHREE)
ranges from 16.50 to 44.39 μg/g, with an average of 30.92 μg/g. The LREE/HREE ratio is 3.09 to
8.77, and $La_N/Yb_N$ = 2.72 to 10.75, with an average of 9.69. These results indicate a degree of
fractionation between LREEs and HREEs, suggesting that the metamorphic protoliths were
sedimentary rocks. Eu anomalies (δEu) range from 0.50 to 0.89, with a mean value of 0.64, and
there are no significant δC (0.55–1.12) anomalies.
Chondrite-normalised LREE patterns (Fig. 6) are right-skewed, while HREE curves are
relatively gentle. All samples show significant negative Eu anomalies, and LREE contents are
much higher than those of chondritic standard values, consistent with the REE distribution
pattern in the khondalite series at the margin of the Yangzi plate. This indicates that the
metamorphic protoliths were sedimentary rocks and claystone; the low-maturity metamorphic
rock series originated from continental crust basement with protoliths composed of sandy and
argillaceous clastic sediments of early Proterozoic immature source areas (Liu XinXin, 2015).
North American shale-normalised REE patterns (Fig. 7) show relatively gentle curves, with
individual samples showing negative Ce anomalies. There are also slightly positive Ho
anomalies, relative LREE enrichment, relative HREE depletion, negative Eu anomalies, and
slightly negative Ce anomalies. Such characteristics are consistent with river, lagoon, and
marginal sea sediments (Deng , 2020 ; Wildman T R et al., 1973; Zhao , 1997; Piper D Z,
1985; Goldstein S J et al., 1988; Murray R W et al., 1991; Sholkovitz E R, Jacobson S B et
al.,1994).

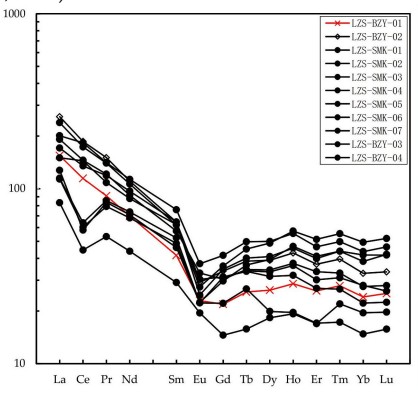

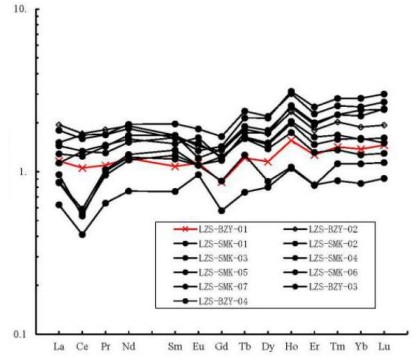

Fig.6 Chondrite-normalized REE patterns for
Lazishao Graphite mine(Boynton W V, 1984)

Fig.7 North American Shale normalized REE
patterns for Lazishao Graphite mine(Haskin L A,
et al., 1968)

### 5.3. Geochemical characteristics of trace elements


Trace element data are shown in Table 4. Large ion lithophile element (LILE) Rb ranges
from 70 to 195 ppm, with an average of 129.0 ppm; Ba ranges from 330 to 850 ppm, with an
average of 564.5 ppm; K ranges from 15,500 to 28,600 ppm, with an average of 22,372.73 ppm; Sr
ranges from 8.80 to 80.8 ppm, with an average of 31.0 ppm. High field strength element (HFSE)
Th varies little, ranging from 6.15 to 18.6ppm, with an average of 10.4 ppm; Nb ranges from 3.34
to 71.0 ppm, with an average of 9.40 ppm; Ta is relatively low, ranging from 6.15 to 18.6 ppm,
with an average of 10.4 ppm; P varies widely from 316.8 to 4048.0 ppm, with an average of
2004.8 ppm; Zr ranges from 141 to 314 ppm, with an average of 223.0 ppm; Hf ranges from 3.06
to 6.50 ppm, with an average of 4.73 ppm; La/Th ranges from 2.84 to 4.71, with a mean value of
3.75. As Sr is relatively enriched in marine sedimentary environments, the Rb/Sr ratio can be
used to distinguish marine and terrestrial sediments (Liang Shuai, 2015). Here, Rb/Sr ranges
from 2.00 to 9.78 (with a mean value of 6.01); all values are > 1, indicating that the sediments are



well sorted and probably originated from a terrestrial depositional environment. Sr/Ba varies in
a relatively small range of 0.02 to 0.13, with a mean value of 0.05.
A spider diagram of trace element ratios relative to original mantle source values (Fig. 8)
shows relative LILE enrichment (e.g., Rb, Ba, and K) and obvious Sr depletion. HFSE, such as
Nb and Ta, show slight depletion, while Zr, Hf, and Th are relatively balanced, with gentle
curves. The content of all elements, except Sr and Ti, are higher than the original mantle source
values.

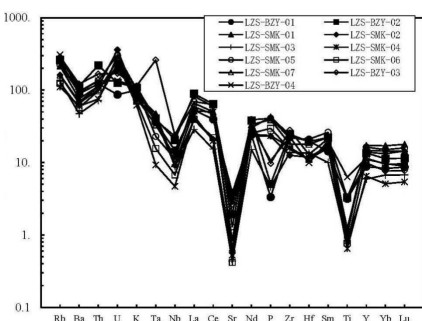

Fig 8 Trace elements primitive mantle standard element cobweb diagram of Lazishao graphite deposit(Sun S S, Mc Donough W F. 1989)

**6. Discussion**
*6.1. Metamorphic protoliths*
The Lazishao graphite deposit has undergone multiple phases of tectonic deformation and
metamorphism, with superposed foliations and mineral assemblages. As such, accurate
recreation of metamorphic protoliths cannot be accomplished merely based on geological
features in the field, mineral assemblages, or mineralogical characteristics; geochemical data are
also needed.
Owing to strong activity of the major components (e.g., $SiO_2$), their contents can change
during multi-phase metamorphism, reducing the accuracy of protolith recreation. Instead,
Winchester et al. (1980) chose relatively inactive elements (Zr, Ti, and Ni) to construct a
$Zr/TiO_2$–Ni diagram. As shown in Fig. 9, data from the four mica–quartz schist samples and
seven graphite ore samples were projected into the zone of sedimentary rocks on a $Zr/TiO_2$–Ni
diagram, suggesting that the metamorphic protoliths of the Lazishao graphite deposit were
sedimentary rocks, and that graphite-bearing metamorphic rocks in the study area are
para-metamorphic.
Simonen (1953) used an Al+fm−C+alk−Si diagram to demonstrate the chemical
characteristics of different metamorphic rocks, and showed wide variations in Al, fm, C, and alk.
Simonen's diagram can effectively eliminate the effects of Si variation on protolith restoration.
Numerous studies have verified that Simonen's diagram performs well in the determining
metamorphic protoliths.
Based on the data in Table 4, Simonen's diagram was plotted for the Lazishao graphite
deposit (Fig. 10). Volcanic rocks plot in the centre, argillaceous sedimentary rocks plot in the
upper left, sandy sedimentary rocks plot in the upper right, and calcareous sedimentary rocks
plot in the lower left. There is no apparent boundary between argillaceous sedimentary and
sandy sedimentary rock zones. All 11 samples projected into the argillaceous sedimentary rock
zone, confirming that the protoliths of the metamorphic rocks in the study area were
sedimentary and that the metamorphic rocks are para-metamorphic.





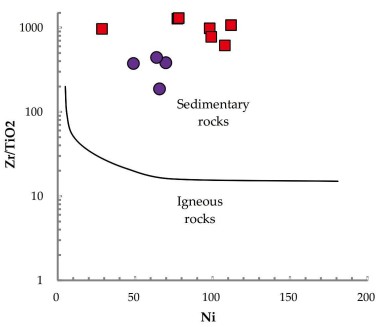

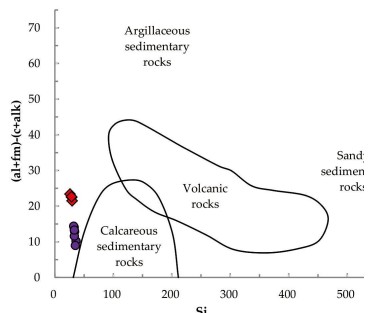

Fig 9 Diagram of Zr/TiO2-Ni of Lazishao graphite deposit(Wang Renmin et al., 1986)

(●are mica–quartz schist samples,■ are graphite ore samples)

Fig 10 Diagram of al+fm-c+alk-Si of Lazishao graphite deposit(Wang Renmin et al., 1986)

(●are graphite ore samples,◆ are mica–quartz schist samples)

221  REEs are incompatible; that is, they cannot enter the crystal structures of rock-forming
minerals or form independent mineral phases. As such, REEs are relatively stable and are not
easily altered by metamorphism or metasomatism, making them suitable for reconstructing
metamorphic protoliths (Meng Hui, 2015). On a La/Yb-$\Sigma$REE diagram (Fig. 11) (Allegre C T,
1978), the 11 samples mostly plot in the shale, claystone, and sandstone zones, providing
further evidence that the protoliths of graphite-bearing metamorphic rocks in the study area
were sedimentary rocks, and that the metamorphic rocks are para-metamorphic.

228  Leake (1969) proposed the (Al–alk)–C diagram to distinguish metasedimentary and
metavolcanic rocks. On an (Al–alk)–C diagram (Fig. 12), the Lazishao graphite samples plot in
the feldspathic claystone and greywacke zones, confirming that the metamorphic protoliths
were feldspathic claystone and greywacke sedimentary rocks.

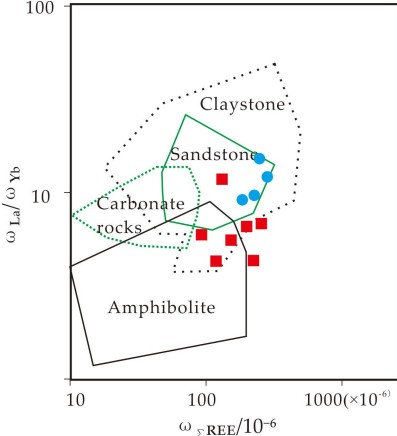

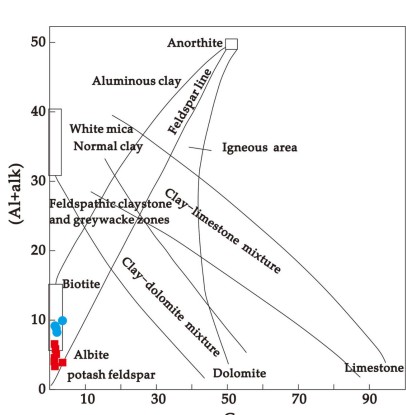

Fig 11 Diagram of La/Yb-$\Sigma$REE of Lazishao graphite deposit(Wang Renmin et al., 1986)

(●are mica–quartz schist samples,■ are graphite ore samples)

Fig 12 Diagram of (al-alk)-c of Lazishao graphite deposit(Wang Renmin et al., 1986)

(●are mica–quartz schist samples,■ are graphite ore samples)






In summary, the protoliths of metamorphic rocks in the Lazishao graphite deposit were
sedimentary rocks, primarily comprised of mud shale and mixed greywacke.
*6.2. Palaeo-sedimentary environment*
Rocks formed in different depositional environments differ in terms of mineral
composition and the contents and ratios of specific elements (Zhao Zhenhua, 1997). Since the
protoliths of the Lazishao graphite deposit were mainly mud shale and greywacke, we infer
that the corresponding depositional environment was a terrestrial or low-energy static shallow
water environment.
The ternary diagram of claystone composition in different climatic zones and the Ba–Sr
diagram proposed by Melezhik and Predovsky (1982) are widely used for distinguishing
claystone depositional environments and palaeo-climatic conditions (Fig. 13). Here, the sample
predominantly plot in the terrestrial facies zone of a cold or moderately cold climate of the
ternary diagram; this is supported by the relatively high $SiO_2$ and $K_2O$, which are indicative of
cold or moderately cold climate. On a Ba–Sr diagram (Fig. 14), almost all sample points plot in
the freshwater environment zone.
In summary, the palaeo-sedimentary environment of the metamorphic protoliths was a
low-salinity terrestrial freshwater body in a cold or moderately cold climatic zone. Combined
with the geochemical characteristics of REEs, we speculate that the sedimentary environment of
the Lazishao graphite deposit was a low-energy static water environment of the fluvial–lagoon
facies.

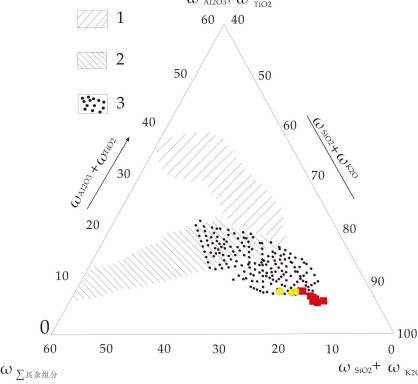

Fig 13 Composition diagram of clay rocks in different climatic zones of Lazishao graphite deposit(Wang Renmin et al., 1986)
1-Terrestrial facies clay compositions in humid and hot climatic zones; 2-Marine facies, lacustrine and lagoon facies clay compositions in dry climatic zones; 3-Terrestrial facies clay compositions in cold or moderately cold climatic zones
( ●are mica–quartz schist samples,■ are graphite ore samples)

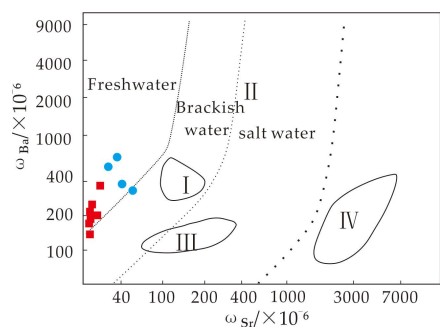

Fig 14 Diagram of Ba-Sr of Lazishao graphite deposit(Wang Renmin et al., 1986)
Ⅰ-Clay in modern deltaic facies brackish water environment; Ⅱ-Pelagic sediments of the Pacific Ocean; Ⅲ-Marine facies carbonate rocks on the Russian platform of different ages; Ⅳ-Modern deposits in a high-salinity waterbody
( ●are mica–quartz schist samples,■ are graphite ore samples)

*6.3. Provenance based on geochemical properties*
Provenance analysis can reveal the location and properties of sediment sources, paths of
sediment transport, and characteristics of sedimentation and tectonic evolution of the basin. The
clastic components and structure of clastic rocks can also directly reflect the tectonic setting of
the provenance area and the sedimentary basin (Liu BaoJun et al .,2006)
The Ni–$TiO_2$ diagram proposed by Floyd et al. (1989) is very accurate in discriminating the
provenance of metamorphic protoliths. On this diagram (Fig. 15), all seven graphite ore samples
plot in the sandstone zone, two of the four mica–quartz schist samples plot in the argillaceous
rock zone, one plots in the felsic rock zone, and one plots between the argillaceous rock and



sandstone zones. Together, this suggests that the provenance of the metamorphic rocks and
graphite ore in the study area is probably argillaceous rock and sandstone.

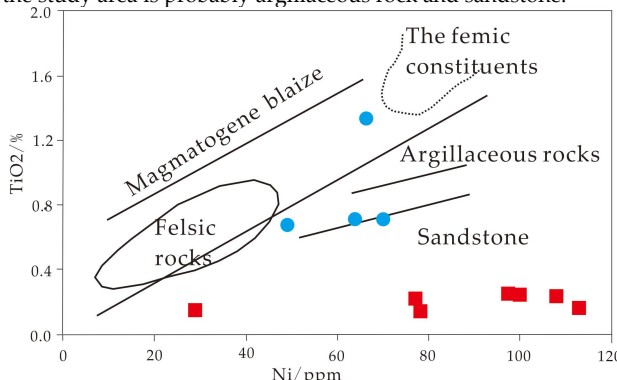

Fig 15    Diagram of Ni-TiO$_2$ of Lazishao graphite deposit
(●are mica–quartz schist samples,■ are graphite ore samples)

The La/Th–Hf diagram (Fig. 16) proposed by Floyd and Leveridge (1987) was adopted to
further verify the provenance of the graphite-bearing metamorphic rocks. All 11 samples plot
within the mixed felsic–intermediate source zone. On the Th–Hf–Co ternary diagram proposed
by Taylor and McLennan (1985) (Fig. 17), all 11 samples plot within the upper crust region.

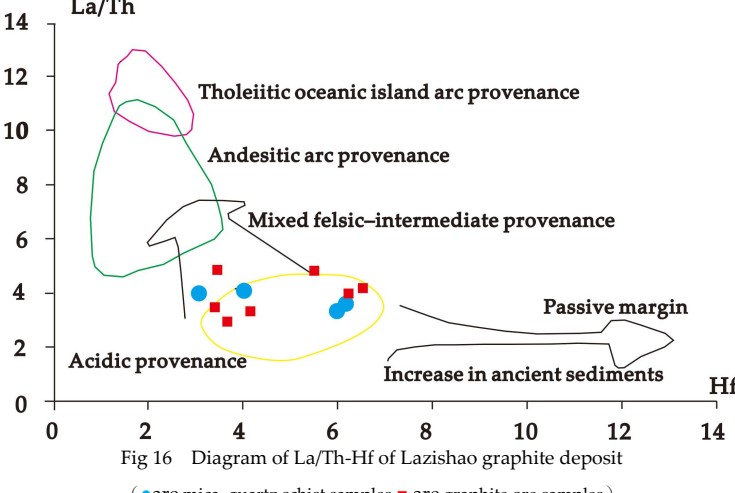

274          Fig 16    Diagram of La/Th-Hf of Lazishao graphite deposit
275          (●are mica–quartz schist samples,■ are graphite ore samples)



Fig 17    Diagram of Th-Hf-Co of Lazishao graphite deposit
(●are mica–quartz schist samples,■ are graphite ore samples)

In summary, the Lazishao graphite deposit originated from the upper crust, with the main
components being argillaceous rock and sandstone from a felsic–intermediate source area.
*6.4 Tectonic environment*
The discrimination formula $Al_2O_3/(Al_2O_3+Fe_2O_3)$ of Jewell and Stallard (1991) is commonly
used to determine geotectonic setting during the deposition of sedimentary rocks. An
$Al_2O_3/(Al_2O_3+Fe_2O_3)$ ratio between 0.6 and 0.9 indicates a continental margin environment, a
ratio between 0.4 and 0.7 indicates a pelagic environment, and a ratio between 0.1 and 0.4
indicates a mid-ocean ridge environment. The $Al_2O_3/(Al_2O_3+Fe_2O_3)$ ratios of the samples from
the Lazishao graphite deposit range from 0.69 to 0.82, with an average of 0.76 (Table 5),
indicating a continental margin environment, which is consistent our other results. Similarly, on
a $K_2O/Na_2O–SiO_2$ diagram [37] (Fig. 18), all seven graphite ore samples and plot in the passive
continental margin region, three of the four mica–quartz schist samples plot in the active
continental margin region, and one of the four mica–quartz schist samples plots in the island
arc region.
Table 5 $Al_2O_3/$($Al_2O_3+Fe_2O_3$)  of Lazishao graphite deposit

| sample number | LZS-BZY-01 | LZS-BZY-02 | LZS-SMK-01 | LZS-SMK-02 | LZS-SMK-03 | LZS-SMK-04 | LZS-SMK-05 | LZS-SMK-06 | LZS-SMK-07 | LZS-BZY-03 | LZS-BZY-04 |
|---|---|---|---|---|---|---|---|---|---|---|---|
| sample type | mica–quartz schist sample | mica–quartz schist sample | graphite ore sample | graphite ore sample | graphite ore sample | graphite ore sample | graphite ore sample | graphite ore sample | graphite ore sample | mica–quartz schist sample | mica–quartz schist sample |
| $Al_2O_3$/$Al_2O_3+Fe_2O_3$ | 0.82 | 0.77 | 0.75 | 0.72 | 0.73 | 0.69 | 0.77 | 0.71 | 0.76 | 0.82 | 0.80 |





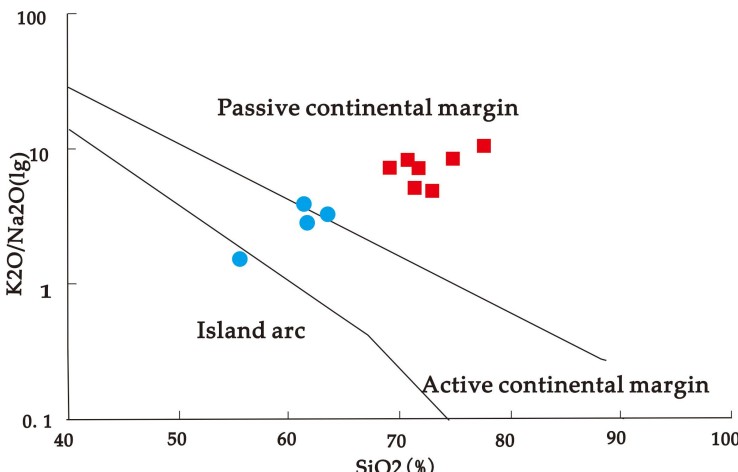

Fig 18 Diagram of K$_2$O/Na$_2$O–SiO$_2$ of Lazishao graphite deposit

(●are mica–quartz schist samples,■ are graphite ore samples)

In summary, protoliths of the metamorphic rocks were deposited along a passive continental margin that remained stable for a sufficient period to form a low-energy freshwater environment in which organic-rich claystone and greywacke were deposited. These deposits were then subjected to regional metamorphism, during which organic carbon was recrystallised into graphite.

## 7. Conclusions

1. The metamorphic rocks of the Lazishao graphite deposit mainly belong to the second member of the Lengzhuguan Formation of the Kangding Group; their lithology is mainly sericite (muscovite)–quartz schist and two-mica–quartz schist.

2. SiO$_2$ in the Lazishao metamorphic rocks is generally high, ranging from 55.60% to 77.94%; K$_2$O > Na$_2$O, and K$_2$O/Na$_2$O+K$_2$O > 0.5; fractionation of LREEs > fractionation of HREEs; there are moderate negative Eu anomalies; ionic lithophile elements (Rb, Ba, and K) are relatively enriched, but Sr is prominently depleted.

3. Lithogeochemical analysis shows that the metamorphic protoliths of the graphite deposit were sedimentary rocks whose lithology was dominated by carbonaceous claystone and greywacke.

4. The palaeo-sedimentary environment was a low-salinity terrestrial freshwater body in a cold or moderately cold climatic zone. Sediments were sourced from the upper crust, and the main provenance components were argillaceous rock and sandstone from a felsic–intermediate source area.

5. Tectonic discrimination diagrams suggest that protoliths of the metamorphic rocks in the study area were probably deposited in an organic-rich fluvial–lagoon facies environment on a continental margin; organic-rich claystone and greywacke were deposited over a long period and then subjected to regional metamorphism during which organic carbon was recrystallised into graphite.





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

**Disclaimer/Publisher's Note:** The statements, opinions and data contained in all publications are solely
those of the individual author(s) and contributor(s) and not of Copernicus Publications and/or the editor(s).
Copernicus Publications and/or the editor(s) disclaim responsibility for any injury to people or property
resulting from any ideas, methods, instructions or products referred to in the content. The contact author
has declared that none of the authors has any competing interests