# Peer review of "Geochemical characterisation and protolith restoration of metamorphic rocks at Lazishao graphite mine, Sichuan"

_EGUsphere, 2023_

## Author Comment (AC2)

October 23(rd), 2023

Chief Executive Editor : Prof. CharLotte Krawczyk

executive editor : Prof. Andrea Di Muro

Topic editor : Prof. Yang Chu

Anonymous Referee #1

*Solid Earth*

Dear Editors and Anonymous Referee #1,

We wish to submit an article for publication in *Solid Earth,* titled "**Geochemical characterisation and protolith restoration of metamorphic rocks at Lazishao graphite mine, Sichuan(EGUSPHERE-2023-1528)**" The paper was coauthored by Haijun Yu, Wenqi Cheng , Xue Wang , Decai Kong , Bo Long .

We are grateful to you and referee for valuable comments that helped us to improve our paper considerably. Hereby we would like to resubmit a revised version of our manuscript where we carefully addressed all points raised by the referee.

We hope that after these changes and thorough revision, our manuscript will be deemed acceptable for publication in *Solid Earth.*

Thank you for your consideration. I look forward to hearing from you.

Sincerely,

Haijun Yu

**Response to comments of Anonymous Referee #1:**

**Specific Comments**
1. **The study is related to one of the huge ore-bearing formation of the graphite deposits. Please provide more detailed discussion on the geochemistry of thegraphite-bearing rocks in the formation, based on the results, implications and conclusions of other previous studies in the same area.**

Reply: Thanks to your suggestions, we have added the following to the "Introduction":In recent years, the Sichuan Bureau of Geology and Mineral Resources conducted mineral surveys on the study area and its surroundings to elucidate the stratigraphy, mineralization-controlling structures, and distribution of igneous rocks. Mineral resource estimations confirmed that the crystalline graphite deposit in the study area comprised a medium-sized deposit. Although previous work provided the geological data necessary for writing this paper, the microscopic structure and deposition characteristics of the graphite deposits have not been studied in great detail, and no systematic work has been done on the deposit's geochemical properties, genesis, and provenance. These gaps in our understanding hinder research on the genesis of graphite deposits surrounding the study area and mineral-prospecting works ( see lines 36-43).
The motivation of this study has been added in the revised version, see lines 44-48.

2. **The formats of some reference need to be corrected.**

Reply: We appreciate it very much for this good suggestion, and we have done it according to your idea (see lines 399-485).

**Small annotation errors**

**Line 12, change "high" to "relatively higher";**

Reply: We appreciate it very much for this good suggestion, and we have done it according to your idea (see line 12).

**Line 25, change "desirable" to "remarkable";**

Reply: We appreciate it very much for this good suggestion, and we have done it according to your idea (see line 22).

**Line 94, change "directional" to "oriented";**

Reply: We appreciate it very much for this good suggestion, and we have done it according to your idea (see line 117).

**Line 97, change "interstitial between" to "interstitial to";**

Reply: We appreciate it very much for this good suggestion, and we have done it according to your idea (see line 119).

**Line 113, change "favourable" to "economically valuable";**

Reply: We appreciate it very much for this good suggestion, and we have done it according to your idea (see line 133).

**Line 122, change "Major elements compositions" to "Major element compositions";**

Reply: We appreciate it very much for this good suggestion, and we have done it according to your idea (see line 147).

**Line 154, change "µg/g" to "ppm";**
Reply: We appreciate it very much for this good suggestion, and we have done it according to your idea (see lines 178-180).

**Line 159, "δC", typo;**
Reply: We are very sorry for our incorrect writing and it is rectified at Line (see line 182).

**Line 217, change "apparent" to "evident";**
Reply: We appreciate it very much for this good suggestion, and we have done it according to your idea (see line 258).

**Fig.1, the legend of Fig.1b is too small;**
Reply: We appreciate it very much for this good suggestion, and we have done it according to your idea (see line 61).

**Fig.5, the labels are too small;**
Reply: We appreciate it very much for this good suggestion, and we have done it according to your idea (see line 175).

**Fig.13, check the typos in labels.**
Reply: We are very sorry for our incorrect writing and it is rectified at Line (see line 310).

[Figure]

**Editing Certificate**

This document certifies that the paper listed below has been edited to ensure that the language is clear and free of errors. The edit was performed by professional editors at Editage, a brand of Cactus Communications. The intent of the author's message was not altered in any way during the editing process. The quality of the edit has been guaranteed, with the assumption that our suggested changes have been accepted and have not been further altered without the knowledge of our editors.

**MANUSCRIPT TITLE**

**Geochemical characterisation and protolith restoration of metamorphic rocks at Lazishao graphite mine, Sichuan**

**AUTHORS**

**Wenqi Cheng 2, Haijun Yu1,2*, Xue Wang 3, Decai Kong 2, Bo Long 2**

**ISSUED ON**

**October 23, 2023**

**JOB CODE**

**KXXZE_1_2**

[Figure]

[Figure]

Prabh Grewal
Senior Vice President - Editage

 helping you get published

Since 2002, Editage has helped over 430,000 authors publish around 1.2 million research papers in scholarly journals across over 1000 disciplines through editorial, translation, transcription, and publication support services. Editage is a brand of Cactus Communications (cactusglobal.com), a science communication and technology company.

[Figure]

**GLOBAL :**
+1(833) 979-0061 | request@editage.com

**CHINA :**
400-120-3020或021-6020-9400 |
fabiao@editage.cn

---

## Author Comment (AC3)

October 23(rd), 2023

Chief Executive Editor : Prof. CharLotte Krawczyk

executive editor : Prof. Andrea Di Muro

Topic editor : Prof. Yang Chu

Anonymous Referee #2

*Solid Earth*

Dear Editors and Anonymous Referee #2,

We wish to submit an article for publication in *Solid Earth,* titled "**Geochemical characterisation and protolith restoration of metamorphic rocks at Lazishao graphite mine, Sichuan(EGUSPHERE-2023-1528)**" The paper was coauthored by Haijun Yu, Wenqi Cheng , Xue Wang , Decai Kong , Bo Long .

We are grateful to you and referee for valuable comments that helped us to improve our paper considerably. Hereby we would like to resubmit a revised version of our manuscript where we carefully addressed all points raised by the referee.

We hope that after these changes and thorough revision, our manuscript will be deemed acceptable for publication in *Solid Earth.*

Thank you for your consideration. I look forward to hearing from you.

Sincerely,

Haijun Yu

**Response to comments of Anonymous Referee #2:**

**Some major points**
**The manuscript is generally well written, while there are still some grammar mistakes, typos and wrong expression. It is suggested to be edited carefully during the revision.**

Reply: Thanks for your suggestion. We have meticulously addressed any grammatical concerns and enlisted the assistance of a language editing service to refine the manuscript. Please see the attached "Editing Certificate" about editing service.

**The motivation of this study is not as clearly stated in the introduction as it should be. The statement ' However, the microscopic characteristics, geochemistry, ore genesis, carbon source, and other deposit features of graphite mines in the region have not been thoroughly explored (lines 40-41) is too simple to evaluate the importance of the present study. At least, a brief introduction about what have been done, what kind of critical issues remaining unresolved and why this work can resolve them is necessary. More important, the issues raised in the introduction are not properly answered by this study.**

Reply: Thanks to your suggestions, we have added the following to the "Introduction":In recent years, the Sichuan Bureau of Geology and Mineral Resources conducted mineral surveys on the study area and its surroundings to elucidate the stratigraphy, mineralization-controlling structures, and distribution of igneous rocks. Mineral resource estimations confirmed that the crystalline graphite deposit in the study area comprised a medium-sized deposit. Although previous work provided the geological data necessary for writing this paper, the microscopic structure and deposition characteristics of the graphite deposits have not been studied in great detail, and no systematic work has been done on the deposit's geochemical properties, genesis, and provenance. These gaps in our understanding hinder research on the genesis of graphite deposits surrounding the study area and mineral-prospecting works ( see lines 36-43).

The motivation of this study has been added in the revised version, see lines 44-48.

**This study simply used some discriminant diagrams based on elemental data to interpret the protolith, sedimentary environment, provenance and tectonic setting for some graphite bearing rocks, while in-depth discussion is deficient. For instance, degree of fractionation between LREEs and HREEs was used to suggest sedimentary protoliths of the studied metamorphic rocks (line 156), while this conclusion is far from convincing.**

Reply:Thanks to your constructive suggestions.
(1) We have made a more deep discussion on the protoliths, depositional environments, material sources, and tectonic settings pertaining to graphitic rocks ( see lines 74-77 , 290-294 , 307-309 ).

(2) We regret that the confusion caused to the reviewer due to the sentence "These findings attest to the discernible fractionation observed between the light and heavy rare earth elements, thereby insinuating the sedimentary rock origins of the metamorphic protoliths."
We kindly beseech the removal of this sentence, as the mere occurrence of LREE and HREE fractionation fails to provide definitive evidence supporting the sedimentary nature of its underlying protolith ( see lines 181-182 ) .

**There are two types of rocks, graphite ores and mica schist, studied. It is better to introduce, discuss and interpret them separately.**

Reply: Thanks to your suggestions, we have made the corresponding changes in the manuscript, as follow:
(1) We have described separately the characteristics that are not identical between the both ( see lines 366-368 ).
(2) We have added the following to subsection 3.2: The mica–quartz schist is grey coloured on its freshly cut surfaces and has a grano-lepidoblastic texture with a schistose structure. Core samples of the schist exhibited a blastobedding structure, and the bedding of the protolith was clear and well-defined, implying that the protolith of these metamorphic rocks is most likely a para-metamorphic rock. The mineral composition consists of: (1) quartz (45%–65%), which is relatively uniformly distributed and has an allotriomorphic-granular texture, a mosaic structure, and particle sizes of 0.1–0.8 mm, (2) muscovite (30%–42%) with a scaly texture, a schistose-oriented structure, and particle sizes of 0.1–0.5 mm (particles <0.1 mm form sericites), and (3) small amounts of pyrite and other secondary minerals (Fig. 2a and Fig. 2b) ( see lines 85-92 ) .
(3) Kindly permit me to maintain the intrinsic framework of the "Discussion".
Since mica-quartz schist is the ore-bearing lithology of graphite ore, the two have similar depositional environments and geochemical properties, we incline to discuss them together, as a means to curtail the unnecessary discourse within the confine of this manuscript.

**Some minor points**
**Line 12: SiO₂ contents of the metamorphic rocks are high....**

Reply:We appreciate it very much for this good suggestion, and we have executed it according to your idea. ( see line 11).

**Line 16: Large ionic lithophile elements (LILE)**

Reply:We appreciate it very much for this good suggestion, and we have executed it according to your idea. ( see line 16).

**Line 23: delete the first sentence**

Reply:We appreciate it very much for this good suggestion, and we have done it according to your idea. ( see line 21).

**Line 49: Yangtze Craton, please check it throughout the text**

Reply:We appreciate it very much for this good suggestion, and we have done it according to your idea. ( see line 51).

**Line 66: have been destroyed, leaving only...**

Reply:We appreciate it very much for this good suggestion, and we have done it according to your idea. ( see lines 71-72).

**Line 101: what does the 'fixed carbon' mean?**

Reply:Fixed carbon refers to the non-volatile components remaining after combustion in graphite samples, primarily comprising graphite carbon and other forms of non-graphitic carbon. The content of fixed carbon increases with the higher purity of graphite.

**Line 115: some more information about where and what is the drill holes maybe helpful.**

Reply: Thanks to your suggestions, and details of the sampling drill core have been added to the manuscript: Eleven fresh samples were collected from drill cores, which were located 200 m to the north of the orebody's centre and were drilled to a depth of 130 m. The roof and floor of the orebody comprised a mica–quartz schist, whereas the graphite deposit was located between 70 and 82 m (making it 12 m thick). Seven graphite-ore cores were obtained from the deposit, two mica–quartz schist samples were obtained from the orebody's roof, and two more mica–quartz schist samples were obtained from the floor. Samples were analysed at the Mineral Testing Centre of Xichang, Sichuan Provincial Bureau of Geology and Mineral Exploration and Development (Table 1) ( see lines 135-141).

**Line 122: for the elemental concentration, large LOI (up to 12) should be considered, especially when the elemental contents are used in the discussion.**

Reply:Thanks to your suggestion. This paper focus on metamorphic mica-quartz schist and graphite ore, which contain some water-bearing minerals, resulting in high LOI. The samples are from drill cores and are not altered by weathering, so the results of the major element analysis reflect the original properties of the samples . We have added the reasons for the large LOI in the manuscript, see lines 167-170.

**Line 128: it will be helpful to explain a little bit more that why this value indicate sedimentary origin. In addition, what is normal sedimentary origin and is there abnormal sedimentary origin?**

Reply: Na is soluble. Dark mineral is less stable than feldspar in sedimentary rocks. Among the feldspar groups, the more stable are potassium feldspar and acidic plagioclase feldspar. According to He Tongxing, Lu Liangzhao, Li Shuxun, et al., 1980. Petrology of metamorphic rocks [M]. Beijing: Geological Publishing House.P168:$K_2O > Na_2O$, and $K_2O/Na_2O + K_2O > 0.5$,indicating that the protoliths of the graphite deposit and mica–quartz schist were of normal sedimentary origin .

**Line 143:what does the sedimentary differentation mean?**

Reply: The diverse products from the weathering of mother rocks, as they journey through the process of transportation, exhibit a distinct arrangement of deposition, dictated by their unique attributes in response to external circumstances. This wondrous occurrence, known as depositional differentiation, bestows a certain order upon their separate settling.

**Line 156: why fractionated LREE and HREE indicates sedimentary protolith?**

Reply: We regret that the confusion caused to the reviewer due to the sentence "These findings attest to the discernible fractionation observed between the light and heavy rare earth elements, thereby insinuating the sedimentary rock origins of the metamorphic protoliths." We kindly beseech the removal of this sentence, as the mere occurrence of LREE and HREE fractionation fails to provide definitive evidence supporting the sedimentary nature of its underlying protolith ( see lines 181-182 ) .

**Line 186: Similarly, why?**

Reply: We regret any confusion caused to the reviewer. Generally, Rb, Ba, and Sr are usually a sound indication of the marine terrestrial environments, with terrestrial depositional environments when Rb/Sr>1 and Sr/Ba<1. References:Cuo Wei et al., 2020 (doi:10.3969/j.issn.1674-9057.2020.03.001);Song Hong et al., 2023(doi: 10.18654/1000-0569/2023.09.09).

The place in the manuscript have been revised to: Here, we found that the Rb/Sr ratio ranged from 2.00 to 9.78 (generally >1, with a mean of 6.01), The Sr/Ba ratio varied over a relatively small range from 0.02 to 0.13, with a mean value of 0.05 , implying that the rocks were well sorted and may have had a terrestrial sedimentary environment (Cuo et al., 2020; Song et al., 2023) ( see lines 219-222 ) .
.

**Line 234: what does the 'mixed greywacke' mean?**
Reply:We are very sorry for our incorrect writing and it is rectified at Line. I've rectified "mixed greywacke" to "greywacke" ( see line 290 ) .

**Fig. 1 incorporate the Hainan and Taiwan islands into the insert figure in Fig. 1a**

Reply:We appreciate it very much for this good suggestion, and we have done it according to your idea ( see line 61 ).

**Figs2-4: combined them into one figure and some representative figures of mica schist are also necessary**

Reply:We appreciate it very much for this good suggestion, and we have done it according to your idea. Two representative figures of mica schist were added (Fig. 2A and Fig. 2B) ( see line 94 ).

**Fig. 5 symbols are too small to be seen clearly**

Reply:We appreciate it very much for this good suggestion, and we have done it according to your idea.  ( see line 175 ).

**Fig. 13 explain how to calculated the ω values either in the text or in the figure caption. Besides, there is Chinese word in the lower left corner**

Reply:We appreciate it very much for this good suggestion, and we have done it according to your ideas. ( see line 310 ).

[Figure]

**Editing Certificate**

This document certifies that the paper listed below has been edited to ensure that the language is clear and free of errors. The edit was performed by professional editors at Editage, a brand of Cactus Communications. The intent of the author's message was not altered in any way during the editing process. The quality of the edit has been guaranteed, with the assumption that our suggested changes have been accepted and have not been further altered without the knowledge of our editors.

**MANUSCRIPT TITLE**

**Geochemical characterisation and protolith restoration of metamorphic rocks at Lazishao graphite mine, Sichuan**

**AUTHORS**

**Wenqi Cheng 2, Haijun Yu1,2*, Xue Wang 3, Decai Kong 2, Bo Long 2**

**ISSUED ON**

**October 23, 2023**

**JOB CODE**

**KXXZE_1_2**

[Figure]

[Figure]

**Prabh Grewal**
**Senior Vice President - Editage**

[Figure]

Since 2002, Editage has helped over 430,000 authors publish around 1.2 million research papers in scholarly journals across over 1000 disciplines through editorial, translation, transcription, and publication support services. Editage is a brand of Cactus Communications (cactusglobal.com), a science communication and technology company.

[Figure]

**GLOBAL :**
+1(833) 979-0061 | request@editage.com

**CHINA :**
400-120-3020或021-6020-9400 |
fabiao@editage.cn